
**Forecasting and Identifying the Meteorological and Hydrological Conditions Favoring the Occurrence of Severe Hazes in Beijing and Shanghai using Deep Learning**

Chien Wang

Laboratoire d'Aerologie, CNRS and University Paul Sabatier
14 Avenue Edouard Belin, 31400 Toulouse, France

March 2021; Revised August 2021

*Correspondence to*: Chien Wang (chien.wang@aero.obs-mip.fr)

**Abstract.** Severe haze or low visibility event caused by abundant atmospheric aerosols has become a serious environmental issue in many countries. A framework based on deep convolutional neural networks containing more than 20 million parameters, namely HazeNet, has been developed to forecast the occurrence of such events in two Asian megacities: Beijing and Shanghai. Trained using time sequential regional maps of up to 16 meteorological and hydrological variables alongside surface visibility data over the past 41 years, the machine has achieved a good overall performance in identifying the haze versus non-haze events and thus their respectively favorite meteorological and hydrological conditions, with a validation accuracy of 80% in both Beijing and Shanghai cases, exceeding the frequency of non-haze events or no-skill forecasting accuracy, and a F1 score specifically for haze events nearly 0.5. Its performance is clearly better during months with high haze frequency, that is all months except dusty April and May in Beijing and from late autumn through entire winter in Shanghai. Certain valuable knowledge has also obtained from the training such as the sensitivity of the machine's performance to the spatial scale of feature patterns that could benefit future applications using meteorological and hydrological data. Furthermore, an unsupervised cluster analysis using features with a greatly reduced dimensionality produced by the trained HazeNet has, arguably for the first time, successfully categorized typical regional meteorological-hydrological regimes alongside local quantities respectively associated with haze and non-haze events in the two targeted cities, providing substantial insights to advance our understandings of this environmental extreme. Interesting similarities in associated weather and hydrological regimes between haze and false alarm clusters, or differences between haze and missing forecasting clusters have also been revealed, implying that factors such as energy consumption variations, long-range aerosol transport, and beyond could also influence the occurrence of hazes, even under unfavorite weather conditions.

## 1 Introduction

Frequent low visibility or haze events caused by elevated abundance of atmospheric aerosols due to fossil fuel and biomass burning have become a serious environmental issue in many Asian countries in recent decades, interrupting economic and societal activities and causing human health issues (*e.g.*, Chan and Yao, 2008; Silva et al., 2013; Lee *et al*., 2017). For example, rapid economic development and urbanization in China have caused various pollution-related health issues particularly in populated metropolitans such as Beijing-Tianjin region and Yangtze River delta centered in Shanghai (*e.g.*, Liu *et al*., 2017). In Singapore, the total economic cost brought by severe hazes in 2015 is estimated to be $510 million (0.17% of the GDP), or $643.5 million based on a wiling-to-pay analysis (Lin *et al*., 2016). To ultimately prevent this detrimental environmental extreme from happening requires rigid emission control measures in place through significant changes in energy consumption as well as land and plantation management. Before all these measures could finally take place, it would be more practical to develop skills to accurately predict the occurrence of hazes hence to allow mitigation measures to be implemented ahead of time.

Severe haze events arise from the solar radiation extinction by aerosols in the atmosphere, this mechanism can be enhanced with the increase of relative humidity that enlarges the size of particles (*e.g.*, Kiehl and Briegleb 1993). Aerosols also need favorite atmospheric transport and mixing conditions to reach places away from their immediate source locations, while their lifetime in the atmosphere can be significantly reduced by rainfall removal. In addition, soil

moisture is also a key to dust emissions. Therefore, meteorological and hydrological conditions
are critical to the occurrence of haze events besides particulate emissions. To forecast the
occurrence of such events using existing atmospheric numerical models developed based on fluid
dynamics and explicit or parameterized representations of physical and chemical processes, the
actual task is to accurately predict the concentration of aerosols at a given geographic location
and a given time in order to correctly derive surface visibility (*e.g.*, Lee *et al.* 2017 & 2018).
However, the propagation of numerical or parameterization errors through the model integration
could easily drift the model away from the original track, not mentioning that lack of real-time
emission data alone would simply handicap such an attempt. Therefore, a more fundamental
issue in practice is whether these models could reproduce the *a posteriori* distribution of the
possible outcomes of the targeted low-probability extreme events. Ultimately, lack of knowledge
about the extreme events would, in turn, hinder the effort to improve the forecasting skills.
Differing from the deterministic models, an alternative statistical prediction approach could
be adopted, if the predictors of a targeted event could be identified and a statistical correlation
between them could be established with confidence. However, this is a rather difficult task for
the traditional approaches, because it requires an analysis dealing with a very large quantity of
high-dimensional data to establish a likely multi-variate and nonlinear correlation that can be
generalized. Nevertheless, such attempts can obviously benefit now from the fast-growing
machine learning (ML) and deep learning (DL) algorithm development (*e.g.*, LeCun *et al.*,
2015). In addition, technological advancement and continuous investment from governments and
other sectors across the world have led to a rapid increase of quantity alongside substantially
improved quality of meteorological, oceanic, hydrological, land, and atmospheric composition
data. These data might still not be sufficient for evaluating and improving certain detailed
aspects of the deterministic forecasting models. However, rich information contained in these
data about favorite environmental conditions for the occurrence of extreme events such as hazes
could already have a great value for developing alternative forecasting skills.
Many Earth science applications dealing with meteorological or hydrological data need a
trained machine to not only forecast values but also recognize patterns or images. However, this
can easily lead to a curse of dimensionality of many traditional ML algorithms. Fortunately, deep
learning that directly links large quantity of raw data with targeted outcomes through deep
convolutional neural networks or CNNs (Goodfellow *et al.*, 2016) offers a clear advantage in
sufficiently training deep networks suitable for solving highly nonlinear issues. In doing so, DL
can also eliminate the possible mistakes in data derivation or selection introduced by subjective
human opinion regarding a poorly understood phenomenon. Recently, DL algorithms have been
explored in various applications in atmospheric, climate, and environmental sciences, ranging
from recognizing specific weather patterns (*e.g.*, Liu *et al.*, 2016; Kurth *et al.*, 2018; Lagerquist
*et al.*, 2019; Chattopadhyay *et al.*, 2020), weather forecasting including hailstorm detection (*e.g.*,
Grover *et al.*, 2015; Shi *et al.*, 2015; Gagne *et al.*, 2019), to deriving model parameterizations
(*e.g.*, Jiang *et al.*, 2018), and beyond.
In certain applications, the targeted outcomes are the same features as the input but at a
different time, *e.g.*, a given weather feature(s) such as temperature or pressure at a given level.
The forecasting can thus be proceeded by using pattern-to-pattern correlation from a sequential
training dataset with spatial-information-preserving full CNNs such as U-net (Ronneberger *et al.*,
2015; Weyn *et al.*, 2020). However, this is certainly not the case for the applications where the
environmental conditions associated with targeted outcome are yet known. For such applications,
a possible solution is to utilize a large quantity of raw data with minimized human intervention in
data selection to train a deep CNN to associate targeted outcomes with favorite environmental
conditions. This study represents such an attempt, where a DL forecast framework is trained to
identify the meteorological and hydrological conditions associated with the occurrences of
severe hazes. The DL framework has been developed initially with the severe hazes in Singapore
(Wang, 2020), and now hazes in two megacities of China, Beijing and Shanghai. In terms of
particulate pollutant emissions, all these cities share certain sources including fossil fuel
combustions from transportation, domestic, and industries. On the other hand, each city also has
its own unique sources, for instance, desert and perhaps anthropogenic dust for Beijing, and
massive biomass burning in Singapore (Chen *et al*., 2013; Liu *et al*., 2017; Lee *et al*., 2017,
2018, & 2019). It is obvious that besides meteorological and hydrological conditions, dynamical
patterns of anthropogenic activities leading to the emissions of particulate matters are also
important factors behind the occurrence of severe hazes. Nevertheless, the major purpose of this
study is to advance our fundamental knowledge about the weather conditions favoring the
occurrence of hazes and, through an in-depth analysis on the forecasting results to identify the
limit of such a machine and thus to provide useful information for establishing a more complete
forecasting platform for the task.
131        In the paper, the architecture alongside method and data for training are firstly described after
this introduction, followed by a discussion of training and validation results. Then, an
unsupervised cluster analysis benefited from the trained machine is introduced along with the
results that furthers the understanding of the CNN's performance and summarizes, for the first
time, the various typical meteorological and hydrological regimes associated with haze versus
non-haze situations in the two cities. The last section concludes the effort and major findings.
**2 Network Architecture, Training Methodology and Data**
**2.1 Network architecture**
139        The convolutional neural network used in this study, the HazeNet (Wang 2020), has been
developed by adopting the general architecture of the CNN developed by the Oxford
University's Visual Geometry Group or VGG-Net (Simonyan and Zisserman, 2015). The actual
structure alongside hyper-parameters of HazeNet have been adjusted and fine-tuned based on
numerous test trainings. In addition, certain techniques that were not available when the original
VGG net was developed, *e.g*., batch normalization (Ioffe and Szegedy, 2015), have been
included as well. The current version for haze applications of Beijing and Shanghai, though
trained separately, contains the same number of parameters of 20,507,161 (11,376 non-trainable)
owing to the same optimized kernel sizes. Figure 1 shows the general architecture of a HazeNet
version with 12 convolutional and 4 dense layers (in total 57 layers).
149        The network has been trained in a standard supervised learning procedure for classification,
where the network takes input features to produce classification output that are then compared
with known results or labels based on observations. The coefficients of the network are thereafter
optimized in order to minimize the error between the prediction and the observation or label. The
loss function used in optimization is cross-entropy (*e.g*., Goodfellow *et al*., 2017). Such a
procedure is repeated until the performance of the network can no longer be improved. In
practice, the trainings usually last about 2000 epochs (each epoch is a training cycle that uses up
the entire training dataset). This procedure in nature is to train a deep CNN to recognize then
associate input features (bundled meteorological and hydrological conditions in this case) with
corresponding class, *i.e*., severe haze events or non-haze events. As a result, the knowledge
specifically about the favorite meteorological and hydrological conditions of severe hazes could
be advanced.

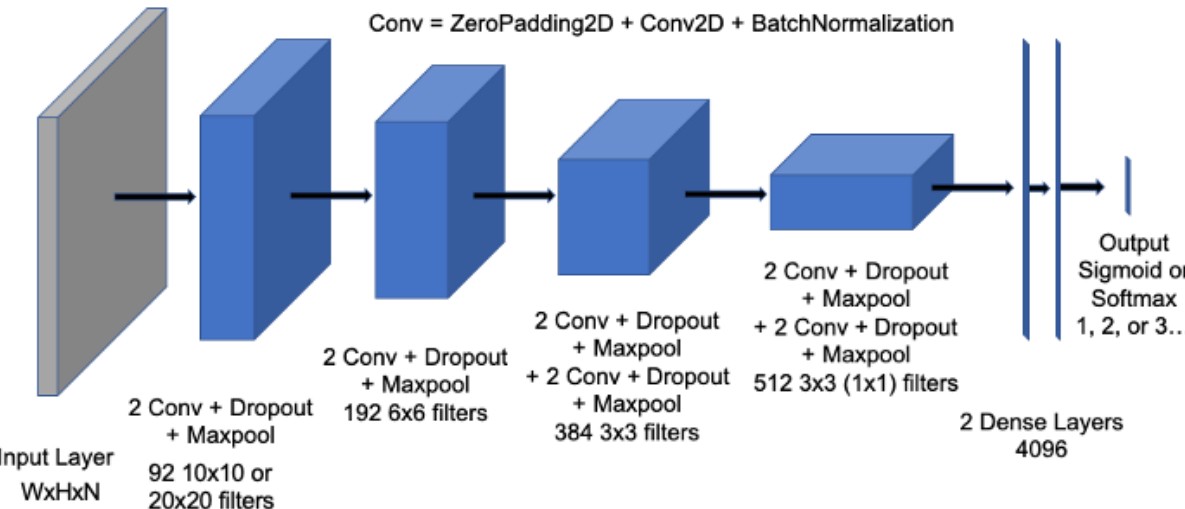

**Figure 1**. Architecture of the 12 convolutional plus 4 dense layer HazeNet. Here "Conv" represents a unit
containing a zero-padding then a 2D convolutional layer, followed by a batch normalization layer. There
is a flatten layer before the 2 dense layers. W = width, H = height, and N = number of features of the
input fields, they are 64, 96, and 16 for Beijing, and 64, 64, and 16 for Shanghai case, respectively.

**2.2 Training data and methodology**

The labels for the training are derived using the observed daily surface visibility (*vis.*
thereafter), obtained from the Global Surface Summary Of the Day or GSOD dataset consisting
of daily observations of meteorological conditions from tens of thousands of airports around the
globe (Smith *et al*., 2011). In the cases of Beijing and Shanghai, data are from observations in
corresponding airports of these two cities during the time from 1979 to 2019, containing 14,975
samples. For simplicity, the discussions will be mainly on the 2-class training, where events with
*vis.* ≤ the long-term mean value of the 25th percentile or p25 of *vis.* (6.27 km in Beijing, 5.95 km
in Shanghai; Fig. 2, right panel; also Fig. S1 in Supplementary) are defined as class 1 or severe
hazes, otherwise the class 0 or non-haze cases. Although p25 values vary interannually, their
long-term means actually represent a substantial reduction of *vis.* due to high particulate
pollution (*e.g.*, Lee *et al*., 2017). Note that unlike in the case of Singapore (Wang 2020), fog and
mist are more common low visibility events in Beijing and Shanghai and thus have been
excluded from the labels of severe hazes by following GSOD fog marks. The number of severe
haze events occurred during 1979-2019 defined in the above procedure is 3099 and 2099 for
Beijing and Shanghai, or in a frequency of 20.7% and 20.0%, respectively.
The training and validation of HazeNet also need the input features with the same sample
dimension of the labels. These input data are derived from hourly maps of meteorological and
hydrological variables covering the data collection domain (Fig. 2, Left), obtained from ERA5
reanalysis data produced by the European Centre for Medium-range Weather Forecasts or
ECMWF (Hersbach *et al*., 2020). These data are distributed in a grid system with a horizontal
spatial interval of 0.25 degree. Up to 16 features are derived from the original hourly data fields
covering the analysis domain respectively for Beijing (64x96 grids) and Shanghai (64x64 grids),
including: daily mean of surface relative humidity (REL thereafter); diurnal change as well as
daily standard deviation of 2-meter temperature or DT2M and T2MS, respectively; daily mean of
10-meter zonal and meridional wind speed or U10 and V10, respectively; daily mean of total
column water (TCW); daily mean (TCV) and diurnal change (DTCV) of total column water
vapor; daily mean of planetary boundary layer height (BLH); daily mean soil water volume in
soil layer 1 and 2 or SW1 and SW2, respectively; daily mean of total cloud cover (TCC); daily
mean geopotential heights at 500 (Z500) and 850 (Z850) hPa pressure levels along with their
diurnal changes D500 and D850, respectively. All input features have been normalized into a
range of [-1, +1] (Fig. S2 in Supplementary).

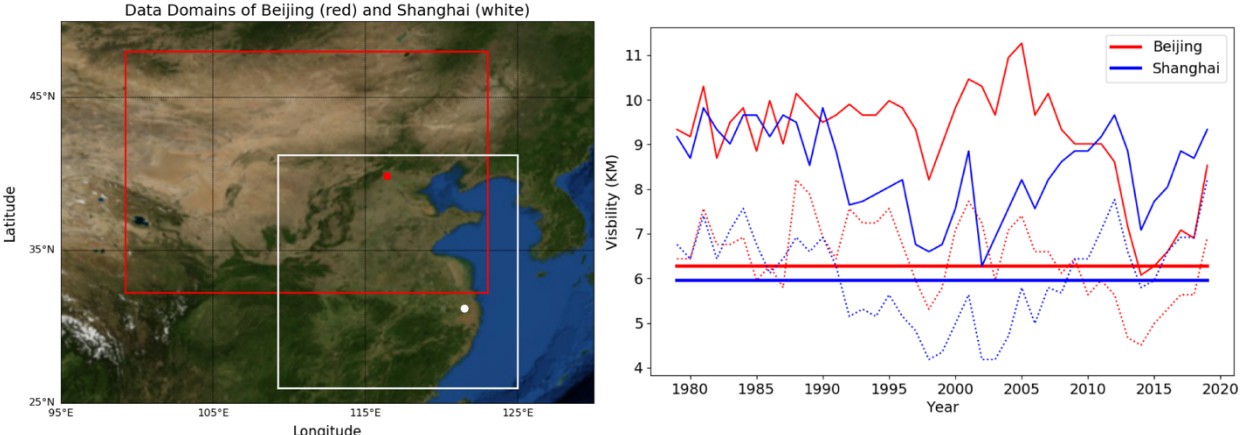

**Figure 2**. (Left) The input-feature defining domains for Beijing (red box and dot, 99.25 - 123E, 32.25-
48N; 96x64 grids with ERA5 data) and Shanghai (white box and dot, 109.25-125E, 26-41.25N; 64x64
grids), made using Basemap library, a matplotlib extension. (Right) Annual means (solid curves), 25[th]
percentiles (dash curves), and 25[th] percentile means (solid straight lines) of surface visibility in Beijing
(red) and Shanghai (blue) between 1979 and 2019.
Before the training, the entire samples of labels alongside corresponding input features were
randomly shuffled first then split as: 2/3 of the samples went to training set and 1/3 to validation
set, each is used duly for its designated purpose throughout the entire training process without
switch. The above procedure treats each of the events as an independent one. For the
convenience in comparing performance or restarting training based on a saved machine, a saved
training dataset alongside a holdout validation dataset that has never been used in training,
produced following the above procedure, was used.
The number of samples used in training HazeNet is rather limited in deep learning standard.
However, to associate 16 joint two-dimensional maps with targeted labels even with the current
number of samples is still a demanding task, requiring a deep CCN to accomplish. Furthermore,
targeted severe hazes are a low probability event. Its frequency of appearance is about 20.0% in
Beijing and Shanghai cases. Therefore, trained machine would easily bias toward the
overwhelming non-haze events. To resolve these issues, a combination of class-weight and batch
normalization has been implemented in HazeNet, both using corresponding Keras functions. The
class weight is to change the weight of training loss of each class, normally by increasing the
weight of the low frequency class. Class weight coefficient was calculated based on the ratio of
class 0 to class 1 frequency. Batch normalization (Ioffe and Szegedy, 2015) is an algorithm to
renormalize the input distribution at certain step (*e.g.*, each mini batch) to eliminate the shift of
such distribution during optimization. The above approach has effectively reduced the overfitting
while overcome the data imbalance issue, making the long training of a deep CNN become
possible (Wang, 2020). Entire trainings have been conducted using a NVIDIA Tesla V100-
SXM2 GPU cluster, costing 25s and 17s per epoch for the machine of Beijing and Shanghai,
respectively.
**2.3 Kernel size optimization**
As in the cases of other CNNs, there are many hyperparameters need to be determined or
optimized. These have been done through numerous testing trainings. In practice, it occurs that,
the deep architecture of HazeNet and the long training procedure have actually made the
performance less sensitive to many hyperparameters of the network. One hyperparameter,
however, is specifically interesting to explore for an application using large quantity of
meteorological maps, that is the kernel size of the first convolutional layer, where the input data,
*i.e.*, meteorological and hydrological maps are convoluted then propagated into the subsequent
layers.

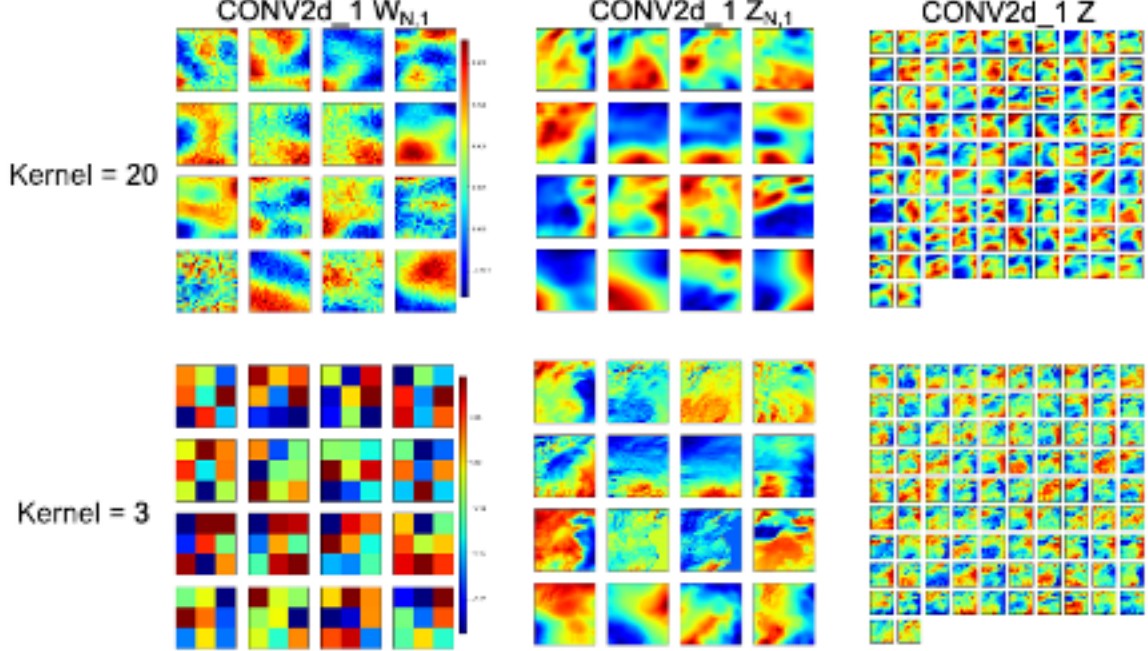

**Figure 3**. (Left column) Weight coefficients of the first filter set ($W_{N,1}$), (Middle column) partial output
for each feature ($Z_{N,1}$), and (Right column) the output ($Z$) of the first convolution layer (CONV2d_1) with
two selected kernel sizes or ks: (upper panels) 20x20 and (lower panels) 3x3. Here $W$ represents the filters
and $Z$ the output of convolution, the subsets of $Z$ before the feature dimension is merged can be expressed
as: $Z_{N,i} = W_{N,i}(ks, ks) \cdot f_N^T(ks, ks),$ with the order of input features $N =1,\dots16$ and $i$ represents the
convolutional layer index, *i.e.*, 1 is the first layer or CONV2d_1. For the first layer, input feature size is
$(h,w) = (64, 64)$, the sets of filters is 92, thus the final output $Z$ has a dimension of $(h\text{-}ks+1, w\text{-}ks+1, 92)$.
Shown are results from the trainings for Shanghai haze cases.
Meteorological maps or images often contain characteristic patterns with different spatial
scales. Intuitively, preserving these patterns could be important in predicting the targeted
extremes. Apparently, a larger kernel size produces smoother output images from the first
convolutional layer, while a smaller kernel size can preserve many spatial details of the
meteorological maps as demonstrated from the layer output shown in Fig. 3. In practice,
however, the patterns produced by the latter configuration might be too complicated for the
networks to recognize and to perform classification, whereas patterns resulted from a relatively
larger kernel size for the first convolutional layer might be more characteristic for the task. The
actual result suggests that HazeNet configured with a first-layer kernel size of 20 to 26 or close
to 5 – 6 degrees in spatial 'resolution', consistently produces a better performance (about a 10%
improvement in *F1 score*) than that by a smaller kernel size of 3 or 6. As a result, a kernel size of
20 has been adopted as the default configuration for the first 2 convolutional layers in this study.

## 3 Training and Validation Results of Haze Forecasting

Currently, it is still difficult to find any practical score in forecasting the occurrence of severe
hazes for comparison. Therefore, the performance of HazeNet has been mainly measured by
using certain commonly adopted metrics for classification largely derived from the concept of
the so-called confusion matrix (*e.g.*, Swets, 1988; Table A), including *accuracy*, *precision*,
*recall*, *F1 score*, *equitable threat score* or *ETS*, and *Heidke skill score* or *HSS* (Appendix A).
Unless otherwise indicated, the discussions on the performance scores are hereafter referring to
the severe haze class, or class 1, and obtained from validation rather than training. In all the
cases, the performance metrics referring to non-haze or class 0 has much better scores. Also note
that, unless otherwise indicated, results shown in this Section are obtained using 16 features.

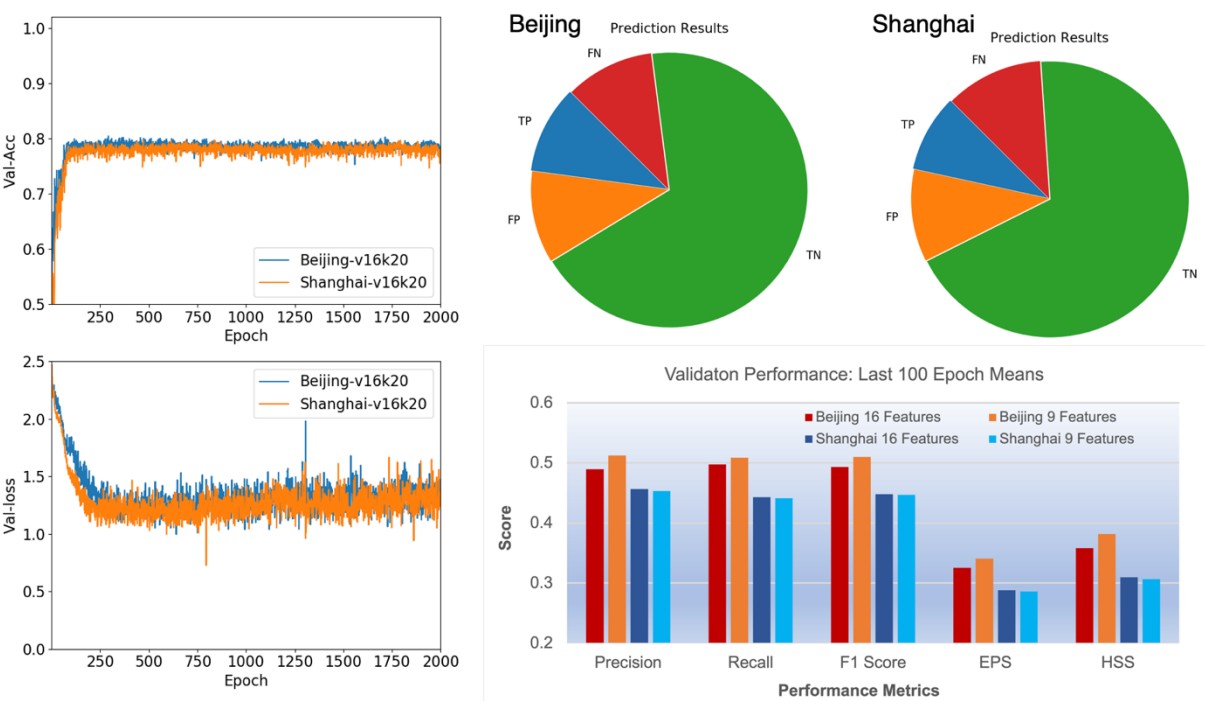

**Figure 4**. (Left) Validation accuracy (top panel) and loss (lower panel) of HazeNet with 16 features for
Beijing and Shanghai cases, kernel size for the first filter is 20x20. (Right Top) Prediction outcomes in
reference to haze events or class 1 of Beijing and Shanghai with 16 features. Here TP = true positive, TN
= true negative, FP = false positive, and FN = false negative prediction outcomes. (Right Bottom) Scores
of performance metrics as last 100 epoch means for Beijing and Shanghai with 16 and 9 features,
respectively.
In order to train a stable machine, trainings with 2000 epochs or longer have been conducted
instead of using certain commonly adopted skills such as early stop. As a result, the validation
performance metrics of the trained machines all appeared to be stabilized by approaching the end
of training (Fig. 4). These scores were consistent with the results of ensemble training with the
same configuration but different randomly selected training and validation datasets, also
comparable among trainings with different configurations. Overfitting has been clearly overcome
due to such a long training procedure alongside the adoption of class weight and batch
normalization. In a 2-class classification (haze *vs*. non-haze), trained deep HazeNet can always
reach an almost perfect training accuracy (*e.g*., 0.9956 for Beijing cases) and a validation
accuracy of 80% (frequency of non-haze events or no-skill forecasting accuracy) in both Beijing
and Shanghai cases (Fig. 4, left). At the same time, the performance scores in predicting
specifically severe hazes are also very reasonable, *e.g*., for Beijing cases either precision or recall
exceeds 0.5 (they normally evolve in opposite direction), leading to a nearly 0.5 *F1 Score* (Fig.4,
right). The corresponding scores in training are obviously much higher, *e.g*., with precision,
recall, and F1 as 0.9804, 0.9980, and 0.9880, respectively for Beijing cases, owing to the deep
and thus powerful CNNs. HazeNet performed slightly better than several known deep CNNs
such as Inception Net V3 (Szegedy *et al*., 2015), ResNet50 (He *et al*., 2015), and VGG-19
(Simonyan and Zisserman, 2015) in the same haze forecasting task (Wang, 2020). Nevertheless,
as indicated previously that a nearly perfect validation performance is not realistic since
meteorological and hydrological conditions are not the only factors behind the occurrence of
haze events.

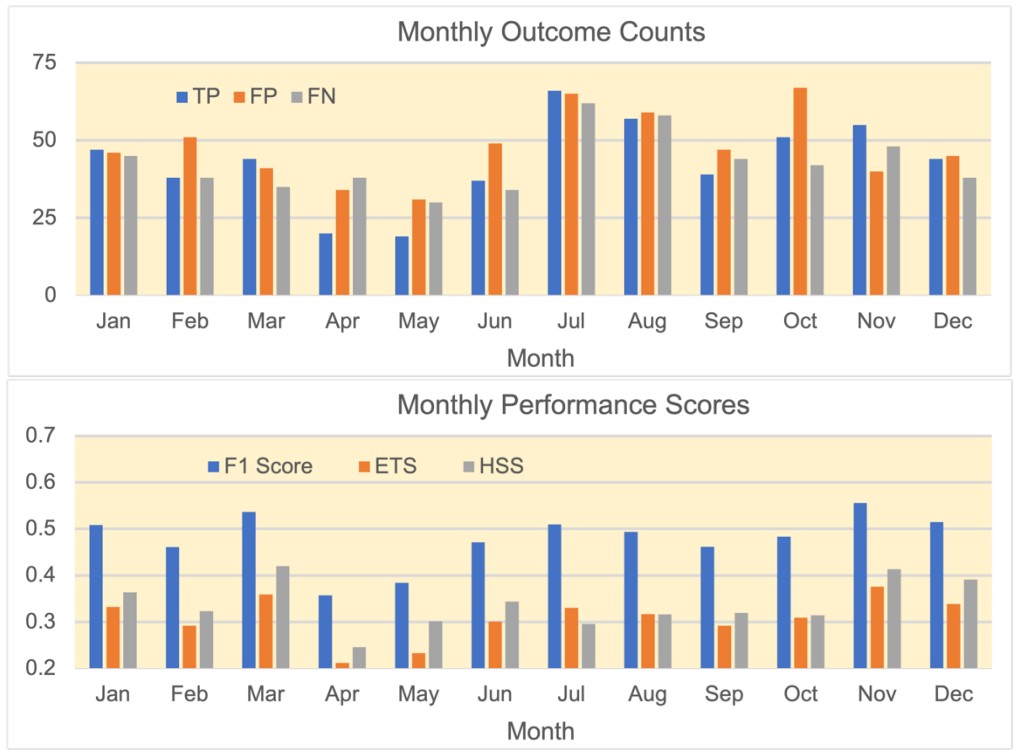

**Figure 5.** (Top) Monthly counts of predicted TP, FP, and FN outcomes and (Bottom) performance scores
for each month. All from validation of Beijing cases with 16 features.
Looking into the specific prediction outcomes in referring to severe haze, the trained machine
has produced considerably higher ratio of true positive or TP outcomes than in the Southeast
Asia cases (Wang, 2020) despite a number of outcomes of false positive or FP (*i.e*., false alarm)
and false negative or FN (*i.e*., missing forecast). In forecasting the severe hazes in Beijing, the
trained machine performs reasonably well throughout all months except for April and May or the
major dusty season there, producing F1 score, ETS, and HSS all exceed or near 0.5 as well as the
number of TP outcomes is higher than that of FN (Fig. 5). HazeNet actually performs better in
months with more observed haze events. For Beijing, the lowest haze season is during the dusty
April and May when all the major performance metrics are lower than 0.4, and the machine
produces more missing forecasts than true positive outcomes. The relatively poor performance in
spring suggests that the weather and hydrological features associated with dust-dominated haze
events during this period might differ from the situations in the other seasons when hazes are
mainly caused by local particulate pollution. For Shanghai cases, HazeNet performs better during
late autumn and entire winter (from November to February) when haze occurs most frequently
(not shown). The worst performance comes from the monsoon season (July to October), or the
season with lowest haze cases.
**Reducing the number of input features**. One recognized advantage of deep CNN in
practice is its capacity to directly link the targeted outcome with a large quantity of raw data,
thus avoid human misjudgment in selecting and abstracting input features due to a lack of
knowledge about the application task. Nevertheless, for an application such as this one that uses
a large number of meteorological and hydrological variables (or channels in machine learning
term), reducing the number of input features with a minimized influence on the performance can
still benefit the efforts of establishing physical or dynamical causal relations and beyond.
There are certain available methods to rank features then reduce some unimportant ones.
These do not work straightforwardly for deep CNNs (*e.g*., McGovern *et al*., 2019). In the
previous effort, this has been done by testing the sensitivity of the full network performance in
real training with either a single feature only or all but one features (Wang, 2020), which
apparently is also a demanding task. Here, another attempt has been made to use a trained then
saved machine to examine the sensitivity of the network to various features (Appendix B).
The sensitivity analyses using trained machines for Beijing and Shanghai have obtained
largely consistent results, indicating that the network is more sensitive to the same 9 features
than the other 7 (Fig. S3). The highest-ranking features though differ, with diurnal change of
column vapor (DTCV) and soil water content in the second soil layer (SW2) as the most
sensitive features for Beijing, while relative humidity (REL) and planetary boundary layer height
(BLH) for Shanghai. Most importantly, trainings using only the top 9 most sensitive features
have produced a performance equivalent to or even better than the same training but with 16
features (Fig. 4, Right Bottom). With reduced number of features, many further analyses can be
conducted with less workload and produce results that are easily understood.
**4 Identifying and Categorizing the Typical Regional Meteorological and Hydrological**
**Regimes Associated with Haze Events**
A major purpose of this study is to identify the meteorological and hydrological conditions
favoring the occurrence of severe hazes in the targeted cities. When using a dataset with a large
number of samples, this type of analyses could be better accomplished by applying, *e.g*., cluster
analysis (*e.g*., Steinhaus, 1957), a standard unsupervised ML algorithm that groups data samples
into various clusters in such a way that samples in the same cluster are more similar to each other
than to those in other clusters. Specifically for this study, the derived clusters would likely
represent various regimes in terms of combined meteorological and hydrological conditions for
associated events. However, applying cluster analysis directly to a large number of samples, each
with a feature volume of ~50000 is an uneasy task. A dimensionality reduction is apparently
needed to reduce the feature volume of data.
In practice, a trained CNN is actually an excellent tool for this purpose. It encodes
(downscales) the input with large feature volume into data with a much smaller size in the so-
called latent space (*i.e.*, the output of the layer before the output layer) while equal predictability
for the targeted events. This functionality of CNN has been used in developing various
generative DL algorithms from variational autoencoder or VAE to different generative
adversarial networks or GANs (*e.g.*, Forest, 2019). Therefore, the trained HazeNet for Beijing
and Shanghai using 9 instead of 16 features, benefited from the effort of reducing the number of
input features as described in the end of last Section, have been used here to produce data with
reduced size suitable for clustering (Fig. 6; see also Appendix C). The new sample-feature set
with a size of 14,975×512 produced from this procedure was then used in cluster analysis.

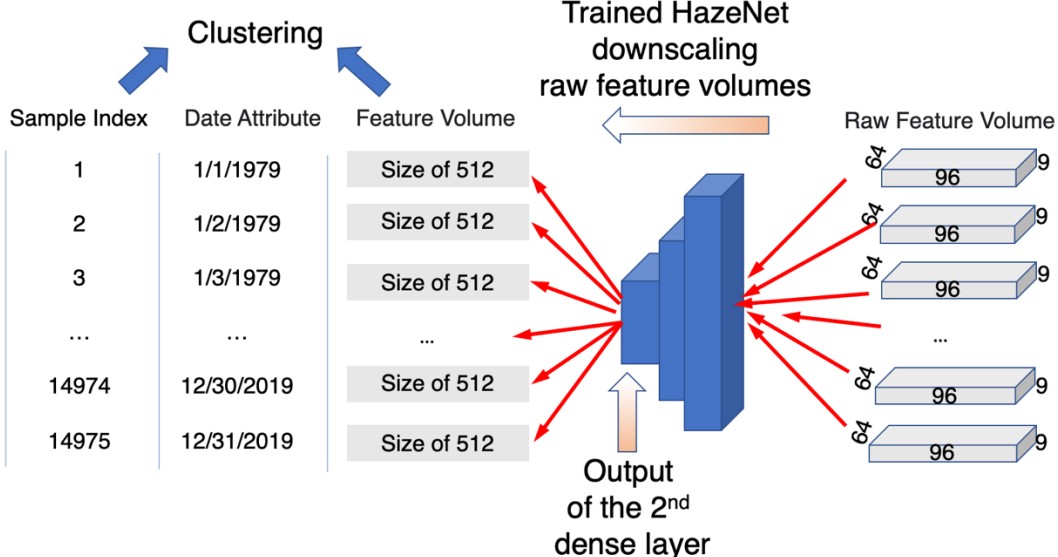

Figure 6. A diagram of the cluster analysis procedure. Here 96, 64, and 9 represent the number of
longitudinal, latitudinal grids, and number of features (variables), or the size of the input feature volume
of a trained HazeNet for Beijing cases, while 512 is the size of the output from the dense layer before
output layer of HazeNet or the new feature volume.
In order to provide useful information for understanding the performance of the trained
networks, the clustering has been performed for each of the prediction outcomes rather than just
haze versus non-haze events (Appendix C). In this configuration, haze associated regimes are
represented by derived clusters of TP plus FN outcomes, while non-haze regimes by those of TN
plus FP. Since the clusters were derived using the indices of samples as the record for members,
the actual feature maps of the members in any cluster thus can be conveniently retrieved then
used to identify the representative regimes in terms of combined 9 meteorological and
hydrological features. Here the clustering results have been analyzed using the feature maps in
both normalized (machine native) and unnormalized (original reanalysis data) format. The
characteristics of various regimes can be easily identified from the former as they represent
anomalies to climatological means. An added benefit is to advance the understanding of the
performance of the trained networks. The analysis using the latter maps aims to better appreciate
the conventional regional and local meteorological and hydrological patterns associated with
various regimes. The feature maps used in both analyses have been averaged across each cluster
for clarity.

## 4.1 Results based on normalized feature maps

As shown in Figure 7, the 4 clusters of true positive or TP in Beijing cases exhibit a clear
similarity in general feature patterns closely surrounding Beijing (marked by a navy dot in the
figure) among themselves.  These common patterns include an isolated small positive relative
humidity (REL) center covering Beijing, associated with mild diurnal variation change (DT2M)
and standard deviation (T2MS) of surface temperature as well as zonal wind (U10), and a lower
boundary layer height (BLH). Weatherwise, Beijing and its immediate surrounding area appear
to be located between two sharply different airmasses occupying respectively the northwestern
and southeastern part of the domain (weather systems usually progress from northwest to
southeast in this region). When relating this to the other feature characteristics, it is likely that
Beijing and nearby area is not experiencing a drastic weather system change such as fronts when
haze occurs, hence the high REL- a critical condition for aerosol to effectively scatter sunlight -
can be easily formed, aided by a stable boundary layer with mild surface wind to allow aerosols
well mix vertically near the ground while without being significantly reduced through advection
diffusion. In addition, relatively high soil water content could fuel the humidity in the air, and
thin while stable low clouds, if exists (judged based on temperature change) could signal a lack
of persistent precipitation. Altogether, these conditions can apparently allow the haze to easily
form, to last, and to effectively scatter sunlight thus reduce visibility. These conditions are also in
a noticeably contrast to those associated with non-haze events represented by TN outcomes (Fig.
S4).
Note that each cluster consists of a collection of 3D data volumes or images, any two clusters
could be sufficiently differentiated should only one of their images differs based on the
clustering derivation algorithm, even though statistically speaking, they very likely belong to the
same population (*i.e*., should be tested statistically). As shown in Fig. 7, the distinctions between
TP clusters are largely reflected from the two different airmasses distant from Beijing, in both
strength and spatial extent particularly from DTCV patterns, likely representing different types
of systems or background regimes. Specifically, a strong DTCV anomalous center seen in cluster
1 and 4 patterns occupies most of the domain west of Beijing and directly influence Beijing and
its nearby area. In contrast, DTCV distributions in cluster 2 and 3 are much weaker, where
Beijing and its immediate neighboring area even appear to be influence more by the southeaster
system. In addition, surface wind distributions of the first two clusters clearly differ from those
of cluster 3 and 4, and the patterns of BLH alongside SW1 and SW2 over Beijing and its
immediate neighboring area of cluster3 also suggests a land-atmosphere exchange condition
differing from that of others. The combinations of these differences across various TP clusters
apparently well defines the various regimes of surrounding weather systems as well as their
influence on Beijing. For TP clusters of Shanghai, the above similarities alongside differences
among various clusters also exist, except where the cluster 1, 2, and 4 maintain more similarities
in feature patterns of distant airmasses from Shanghai, while cluster 3 offers certain evident
diversity in many feature patterns comparing to other clusters (Fig. S5). Even more interestingly,
the distribution of the number of members within various TP clusters does not differ evidently in
different months (Table S1) (note that the number of haze events itself differs seasonally – Fig.
5). Therefore, it is very likely that the characteristic weather conditions favoring haze occurrence
and being captured by HazeNet cannot be simply differentiated by locations (Beijing vs.
Shanghai) and seasons.

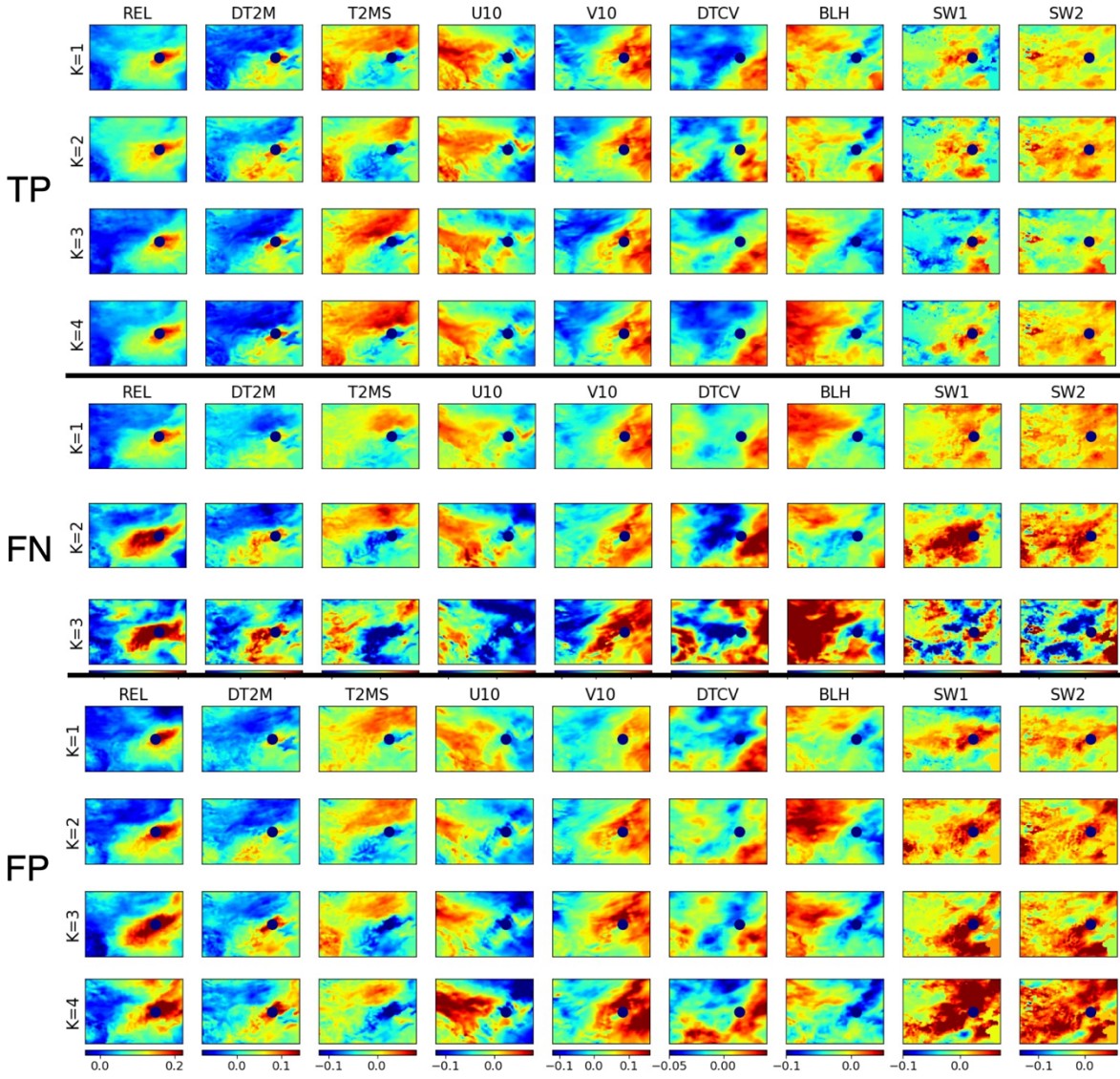

**Figure 7**. Maps of 9 features in normalized format for 4 clusters of true positive or TP outcome, 3 clusters
of false negative or FN outcome, and 4 clusters of false positive or FP outcome. Here TP plus FN = haze
events. Results shown are cluster averages for Beijing (location marked by navy dot) cases.
On the other hand, among three FN clusters (also associated with haze events but missed in
prediction), only the first cluster (the major cluster of FN) displays certain similarity to TP
clusters across various features. Even for this cluster, the characters of the airmasses distantly
surrounding Beijing differ substantially from those of TP clusters, as seen from the patterns of
temperature (DT2M, T2SM), wind particularly V10, and column water (DTCV) that reflect a
much weaker weather system on the west. The patterns of BLH, SW1, and SW2 also differ from
those of TP, indicating a different near site boundary layer and hydrological condition. Such
differences appear to be even more evidently in the two other (minor) clusters, *e.g.*, the size and
strength of high relative humidity center covering Beijing are even different. This result suggests
a possible reason for why HazeNet missed these targets, that is haze might occur under
unfavorable weather and hydrological conditions owing to, *e.g.*, certain energy consumption
scenarios. Again, the distribution of members of these latter two clusters does not exhibit clear
seasonality (Table S1). Interestingly, first two of the four FP (false alarm) clusters display more
clear similarity in normalized feature patterns to those of TP than FN in Beijing and its
immediate surrounding area (Fig. 7). As in FN cases, however, two other clusters differ more
evidently. All these could explain the false alarming made by the machine, *i.e.*, the machine
could have simply been confused by such similarities between certain TP and FP members.
Nevertheless, these could also suggest an alternative reason behind the incorrect forecasts that is
certain pollution mitigation measures were in place. The results of FP clusters and the last FN
cluster besides TP of Shanghai cases also share some similar characters as analyzed here (Fig S5
& S6).
Therefore, it is worth indicating again that meteorological or hydrological conditions are not
the only factors determining the occurrence of hazes. Other factors such as abnormal energy
consumption events or long-range transport of aerosols could all cause haze to occur even under
unfavorable weather and hydrological conditions. This could well be the reason for some of the
missing forecasts (FN outcomes) when haze occurred under unfavorable conditions, as suggested
above, or for false alarms (FP outcomes) when low aerosol events occurred even under a weather
condition favorable to haze. Future improvement of the skill could benefit from this knowledge.
**4.2 Results based on original unnormalized feature maps**
Utilizing feature maps in their original unnormalized format represented by actual physical
quantities could provide a convenience to appreciate the conventional regional and local
meteorological and hydrological patterns, and to detect thus to implement additional analysis, if
necessary, on the possible impact of seasonality or trend associated with various events. Note
that the visual differences between unnormalized feature maps particularly in cluster-mean
format might be subtle for bare eyes to recognize.
For haze events in Beijing (*i.e.*, TP and FN outcomes; Fig. 8), the associated cluster-mean
regional meteorological and hydrological patterns of most features except DTCV contain two
regions with sharply contrasting quantities, roughly separated by a line linking the southwest and
northeast corner of the domain, likely due to the typical progression direction of weather systems
in this region besides meridional variation of general climate. In comparison, as same as shown
in the previous analysis using normalized feature maps, the patterns of the first FN cluster share
many characters with those of TP clusters. The differences among TP and FN clusters are more
evident in DTCV (specifically cluster 1 and 4 versus cluster 2 and 3), SW1, SW2, and surface
winds particularly for the 2$^{nd}$ and 3$^{rd}$ FN clusters. FP clusters also display a similarity to those of
TP clusters (Fig. S7), whereas TN clusters show more visible differences particularly in patterns
of meridional wind (V10) and daily change of column water vapor or DTCV (Fig. S8).
The general regional meteorological and hydrological conditions during haze events in the
southeastern in comparison to the northwestern portion of the domain include a higher relative
humidity, lower variation of surface temperature, largely northward or northwestward wind,
lower planetary boundary layer height, and higher soil water content, and quantity wise these are
all in a sharp contrast to the situations in the other half of the domain. Based on the surface wind
direction, Beijing and its immediate surrounding area is clearly located between two airmasses
both with anticyclonic surface winds. The strengths of these two centers differ particularly in the
last two FN clusters, implying regimes with systems having different strengths or in different
development phases. Such a difference is also clearly related to the visually recognized cross-
cluster difference in DTCV patterns, represented by a strong negative center in the middle of the
domain with varying extent and strength across different clusters. Consistent to the analysis
result using normalized feature maps, all these indicate a stable weather condition over Beijing
and its neighboring area during haze events while surrounded by two (or more) different weather
systems. It is known that dust can cause low visibility events in Beijing. During dust seasons, the
condition of the northwestern half of the domain, represented by a dominant eastward wind and
lower soil water content likely favors dust transport from desert to Beijing. However, the details
would need an in-depth analysis to examine since most clusters having members rather well
distributed through different months (Table S1).

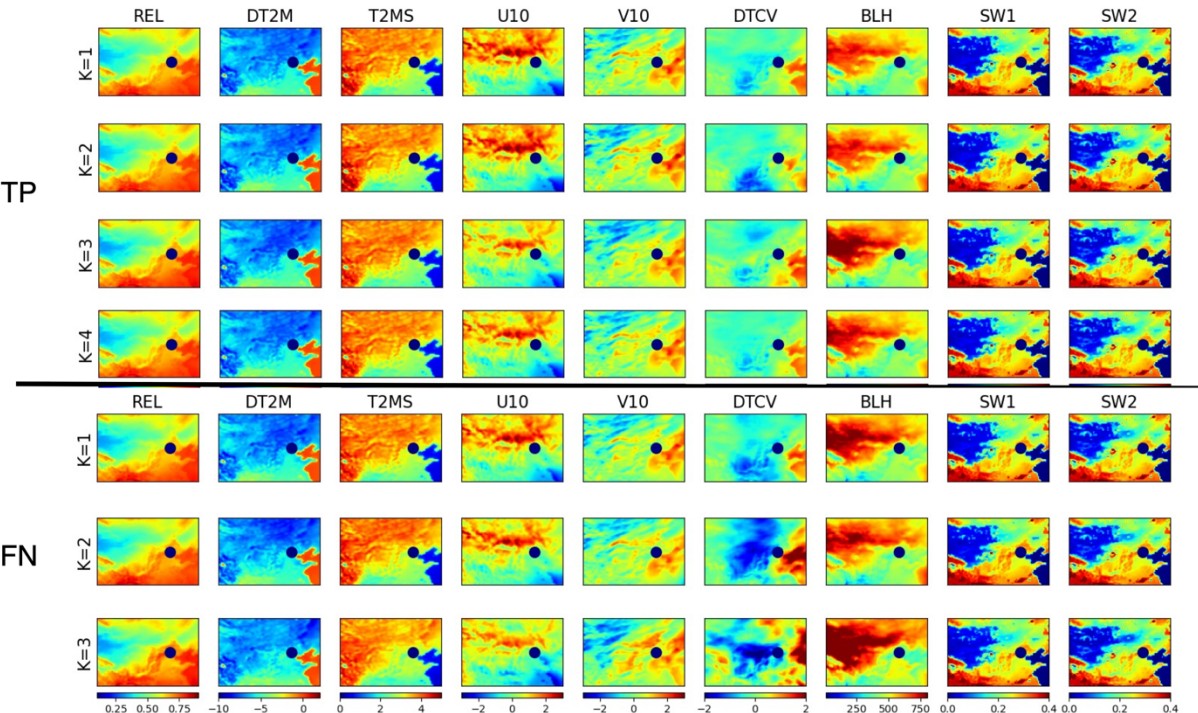

**Figure 8**. Feature maps associated with severe haze events in Beijing represented by 4 clusters of TP (4
top rows) and 3 clusters of FN (3 lower rows) predicted outcomes. Shown are cluster means of
unnormalized data of relative humidity or REL (ratio), diurnal change (DT2M) and daily standard
deviation (T2MS) of 2-meter temperature in degree, 10-meter winds U10 and V10 in m/s, diurnal change
of column water vapor or DTCV $(kg/m^2)$, planetary boundary hheight ot BLH in meter, and soil water
content in soil level 1 (SW1) and level 2 (SW2) in $kg/m^2$.

502        The cluster-means of 9 features for haze events (TP plus FN) versus non-haze (TN plus FP)
at the grid point of Beijing are also derived and listed in Table 1 for reference. Specifically, the
common local conditions associated with hazes in Beijing in comparison to those with non-haze
events include a higher humidity, less drastic variations in surface temperature, a northwestward
rather than southeastward wind, a lower planetary boundary layer height, and higher soil water
contents. Again, the most recognizable cross-cluster differences appear in DTCV (*i.e.*, cluster 1
versus others), followed by surface wind (cluster 1 and 2 versus 3 and 4). In most of the local
features, variabilities of FN clusters tend to be larger than those of TP clusters. Notably, such
differences in local feature quantities for FN clusters are not necessarily more evident than in the
regional maps over distant airmasses. One interesting result of the local weather conditions
shown in Table 1 is that the cluster means of TN are sharply different than those of TP and FN,
while the cluster means of FP and those of TP+FN are likely to be statistically indifferent except

for DTCV, providing an evidence to support the assumption that FP outcomes might simply represent the non-haze events caused by reasons other than weather and hydrological conditions.

For the case of Shanghai, the general weather conditions associated with haze events are likely stable, with characters similar to the cases of Beijing except for that Shanghai appears to be located between a northwest airmass with anticyclonic surface wind and a southeast one with cyclonic wind (Fig. 9). Quantities of most feature patterns display a sharply southeast versus northwest contrast. DTCV maps display a negative center over a large area, its distribution and extent vary significantly among different clusters in particular for the first two FN clusters. The patterns of soil water content in both soil layers exhibit a sharp meridional contrast, much higher in the south part of the domain than in the north part, largely separated by the Yellow River. Local quantities of all the features associated with haze events (TP plus FN) in Shanghai display clear differences with those of non-haze prediction outcomes (TN) (Table 1). The most recognizable cross-cluster differences for TP appear in U10 of cluster 4 and V10 of cluster 3, differing from the cases of Beijing, and DTCV particularly of cluster 3 for FN. Like the cases of Beijing, the cluster mean of the FP outcomes is statistically indifferent to those of haze (TP and FN) than predicted non-haze (TN) events. Again, this result implies that even a weather pattern favoring haze appeared and was correctly recognized by HazeNet, due to other factors such as energy consumption variations, haze could still not to occur.

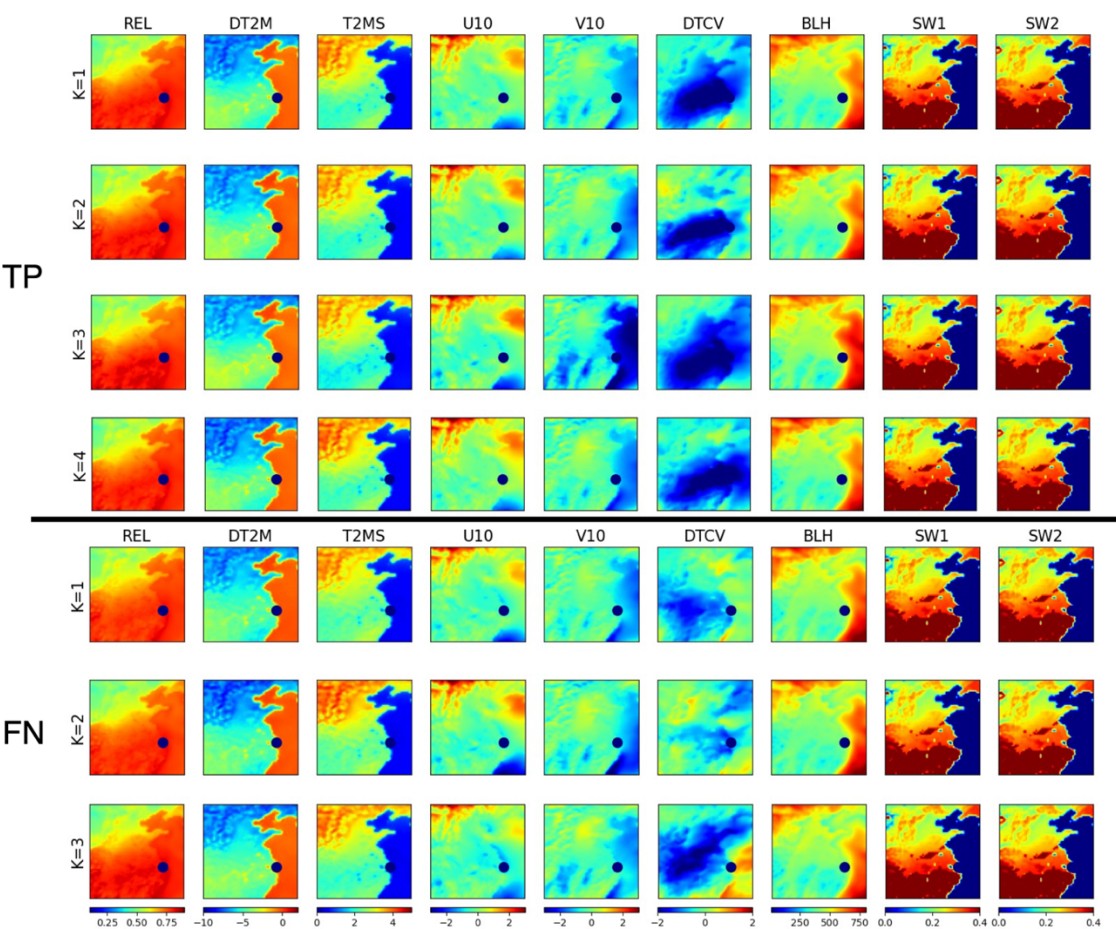

**Figure 9.** The same as Figure 9 except for Shanghai with 4 clusters for TP and 3 for FN outcomes.

It is worth indicating that the current analysis discussed here is only applied to the included
features in clustering, and the presented figures in cluster-wise averaging format might have
effectively smoothed out certain variability among members. A full-scale analysis would
necessarily go beyond this to provide further synoptical or large-scale hydrological insights and
better define different regimes.
**Table 1**. Cluster means of features associated with haze events (TP and FN) in Beijing and Shanghai
versus means of all clusters of non-haze events of TN and FP, respectively. Number of cluster members
of each cluster are listed in bracket.

| Cluster | REL (0-1) | DT2 (°C) | T2MS (°C) | U10 (m/s) | V10 (m/s) | DTCV (kg/m²) | BLH (m) | SW1 (kg/m²) | SW2 (kg/m²) |
|---|---|---|---|---|---|---|---|---|---|
| Beijing | | | | | | | | | |
| TP1 (848) | 0.64 | -5.99 | 3.24 | -0.29 | 0.20 | 0.04 | 379.71 | 0.23 | 0.22 |
| TP2 (181) | 0.65 | -5.80 | 3.14 | -0.28 | 0.19 | 0.57 | 378.33 | 0.23 | 0.23 |
| TP3 (354) | 0.65 | -5.39 | 2.98 | -0.45 | 0.29 | 0.31 | 400.20 | 0.23 | 0.22 |
| TP4 (1208) | 0.64 | -5.82 | 3.18 | -0.34 | 0.28 | 0.27 | 381.28 | 0.23 | 0.22 |
| FN1 (392) | 0.63 | -6.24 | 3.32 | -0.25 | 0.20 | 0.07 | 422.60 | 0.23 | 0.22 |
| FN2 (90) | 0.65 | -5.71 | 3.05 | -0.20 | 0.17 | 0.19 | 406.65 | 0.23 | 0.22 |
| FN3 (26) | 0.69 | -5.37 | 2.94 | -0.61 | 0.39 | -0.17 | 410.95 | 0.25 | 0.23 |
| TN mean | 0.51 | -7.13 | 3.65 | 0.15 | -0.15 | 0.36 | 552.90 | 0.22 | 0.21 |
| FP mean | 0.65 | -5.84 | 3.15 | -0.35 | 0.25 | -0.26 | 386.27 | 0.24 | 0.23 |
| Shanghai | | | | | | | | | |
| TP1 (1228) | 0.81 | -3.44 | 1.79 | -0.16 | -0.55 | -2.25 | 415.59 | 0.35 | 0.35 |
| TP2 (135) | 0.81 | -3.10 | 1.71 | -0.12 | -0.66 | -2.08 | 422.04 | 0.36 | 0.36 |
| TP3 (689) | 0.81 | -2.95 | 1.59 | -0.17 | -1.28 | -2.29 | 472.74 | 0.36 | 0.35 |
| TP4 (355) | 0.81 | -3.52 | 1.82 | 0.03 | -0.57 | -2.74 | 411.96 | 0.35 | 0.35 |
| FN1 (372) | 0.80 | -3.48 | 1.80 | -0.41 | -0.42 | -0.84 | 421.13 | 0.35 | 0.35 |
| FN2 (113) | 0.80 | -3.64 | 1.84 | -0.34 | -0.51 | -1.21 | 423.09 | 0.35 | 0.34 |
| FN3 (107) | 0.82 | -3.28 | 1.77 | -0.68 | -0.49 | 0.10 | 422.36 | 0.35 | 0.35 |
| TN mean | 0.77 | -3.29 | 1.57 | -2.86 | 1.40 | 0.62 | 739.75 | 0.31 | 0.32 |
| FP mean | 0.82 | -3.26 | 1.71 | -0.48 | -0.85 | -2.26 | 438.55 | 0.35 | 0.35 |

## 5 Summary and Conclusions

Following an earlier preliminary attempt for hazes in Singapore, a deep convolutional neural
network containing more than 20 million parameters, namely HazeNet, has been further
developed to test forecasting the occurrence of severe haze events during 1979-2019 in two
metropolitans of Asia, Beijing and Shanghai. By training the machine to recognize regional
patterns of meteorological and hydrological features associated with haze events, the study
would advance our knowledge about this still poorly known environmental extreme. The deep
CNN has been trained in a supervised learning procedure using the time sequential maps of up to
16 meteorological and hydrological variables or features as inputs and surface visibility
observations as the labels.
Even with a rather limited samples (14,975), the trained machine has displayed a reasonable
performance measured by commonly adopted validation metrics. Its performance is clearly better
during months with high haze frequency, *i.e.*, all months except dusty April and May in Beijing
and from late autumn through entire winter in Shanghai. Relatively larger spatial patterns appear
to be more effective than the smaller ones to influence the performance of forecasting. On the
other hand, in-depth analysis on performance results has also indicated certain limitations of
current approach of solely using meteorological and hydrological data in performing forecast.
The trained machine has also been used to examine the sensitivity of the CNN to various
input features and thus to identify then remove features ineffective to the performance of the
machine. In addition, to further categorize typical regional weather and hydrological patterns
associated with severe haze versus non-haze events, an unsupervised cluster analysis has been
subsequently conducted, benefited from using features with greatly reduced dimensionality
produced by the trained machine.
The cluster analysis has, arguably for the first time, successfully categorized major regional
meteorological and hydrological patterns associated with severe haze and non-haze events in
Beijing and Shanghai into a limited number of representative groups, with the typical feature
patterns of these clustered groups derived. It has been found that the typical weather and
hydrological regimes of haze events in Beijing and Shanghai are rather stable conditions,
represented by anomalously high relative humidity, low planetary boundary layer height, mild
daily temperature change that likely associated with a thin low cloud cover over the haze
occurring regions. The result has further revealed a rather strong similarities in associated
meteorological and hydrological regimes between haze and false alarm clusters, or differences
between haze and missing forecasting clusters, implying that factors such as energy consumption
variations, long range transport of aerosols, or beyond, could influence the occurrence of hazes
even under unfavorite weather conditions.
Due to the exploratory nature of this specific effort, several aspects could be further
optimized including the rather arbitrary though statistically meaningful labeling. Also, an in-
depth analysis on weather regimes would necessarily involve the use of certain features that are
not included in the current clustering, which, however, exceeds the extent of this paper and can
only be discussed properly in a future work. Nevertheless, this study has demonstrated the
potential of applying deep CNNs with extensive multi-dimensional and time sequential
environmental images to advance our understandings about poorly known environmental and
weather extremes. The methodology, results alongside experience obtained from this study could
benefit future improvement of the skills. Besides, the trained machines can be used in many
other types of machine learning and deep learning applications as partially demonstrated here.
**Appendix A. Performance metrics**
Several commonly used performance metrics have been used in this study. They are largely derived based on
the so-called confusion matrix (e.g., Swets, 1988) as defined in the following Table A.
**Table A**. Confusion matrix for measuring the prediction outcomes of a given class.

| | | *Observed* | |
|---|---|---|---|
| | | *Positive* | *Negative* |
| *Predicted* | *Positive* | True Positive or TP | False Positive or FP |
| | *Negative* | False Negative or FN | True Negative or TN |

Here, *positive* or *negative* is referring to the outcome of a given event or class in the classification, *e.g.*, severe haze
or non-haze events. Hence, the prediction outcome TP is a correct forecast of a severe haze while TN a correct
forecast of a non-haze event, FP represents a false alarm, and FN a missing forecast. The context of outcomes
changes when the designated class is switched. The major performance metrics used in this paper include:
$$accuracy = \frac{TP+TN}{N} \qquad\qquad (A1)$$
$$precision = \frac{TP}{TP+FP} \qquad\qquad (A2)$$
$$recall = \frac{TP}{TP+FN} \qquad\qquad (A3)$$
$$F1\ score = 2 \cdot \frac{precision \cdot recall}{precision+recall} \qquad\qquad (A4)$$
$$ETS = \frac{TP-Hit_{random}}{TP+FP+FN-Hit_{random}}; \qquad\qquad (A5a)$$
$$where: \quad Hit_{random} = \frac{(TP+FN)\cdot(TP+FP)}{N} \qquad\qquad (A5b)$$
$$HSS = \frac{2\cdot(TP\cdot TN-FP\cdot FN)}{(TP+FP)\cdot(FP+TN)+(TP+FN)\cdot(TP+TN)} \qquad\qquad (A6)$$
Note that *accuracy* has the same value for all the classes and thus is a good metrics for the overall classification.
Values of all the other metrics differ depending on the referred specific class. Here, *F1 score* is the F-score with $\beta =$
1 (van Rijsbergen, 1974), *ETS* represents equitable threat score (or Gilbert skill score; Gilbert, 1884; range = [-1/3,
1]), *HSS* represents Heidke skill score (Heidke, 1926; range = [-∞,1]), and *N* is the number of total outcomes.

## 609 Appendix B. Examining the network's sensitivity to features using trained machine

A method has been adopted in this study to use a trained machine from basic training to examine the sensitivity
of the network to a random perturbation applied to the values of different features. The saved machine contains all
the coefficients in different network layers and can be used to predict output from any of these layers using same
input features for training or validation. The sensitivity of the network to a given feature is determined by comparing
the prediction using input feature maps containing randomly perturbation applied to the map of this feature with the
prediction using original input feature maps, and measured by the content loss between these two predictions, with
*img1* with *MxN* pixels as the unperturbed and *img2* as perturbed network output:
$$Content\ Loss = \frac{1}{M\times N}\sum_{i,j}^{M,N}(img1_{i,j} - img2_{i,j})^2 \qquad\qquad (B1)$$
The perturbation is applied as random patch with addition of -0.2 or 0.2 to 10% of the pixels of the input map of
the targeted feature in each sample while maps of all the other features remain unperturbed. To reduce the workload,
only validation input set corresponding to the class 1 events (about 1020 samples) are used. Therefore, the
sensitivity tested here is actually the sensitivity of the network to a given feature in predicting class 1 events. To
preserve the spatial information of the perturbation field, the output of the 9th layer, or the MaxPooling layer
following the second convolutional layer (Fig. 1) is used as the prediction. It has a size of (15, 31, 92) for Beijing
cases and (15, 15, 92) for Shanghai cases when a kernel size of 20x20 is adopted. A higher content loss represents
that the performance of the network is more sensitive to the variations in value of this feature.

## 626 Appendix C. Cluster analysis

The cluster analysis of this study was conducted in the following three steps (see also Fig. 6).
**(i)** Firstly, the trained and saved HazeNet for both Beijing and Shanghai cases with 9 input features have been
used to perform prediction using the entire 14,975 input samples in original raw data format, *i.e.*, with a feature
volume size of 96x64x9 for Beijing and 64x64x9 for Shanghai for each sample. The prediction results were then
summarized into various outcomes, *e.g.,* as true positive (TP), true negative (TN), false positive (FP), or false
negative (FN) in referring to the haze class. In the meantime, the output of the second dense layer just before the
output layer or latent space (see Fig. 1 & Fig. 6) were further used to form the new data of each sample with reduced
feature volume of 512. This new dataset with 14075 samples and 512 feature volume were ready for clustering.
**(ii)** The second step is to actually perform clustering using the new data with reduced size resulted from the
previous step. For this purpose, it should be conducted separately for different types of samples or events, *e.g.*,
categorizing all the samples for haze into characteristic groups with similarity and same for non-haze events. In
order to provide additional information to further the understanding of the network's performance, the clustering
was actually conducted for different prediction outcomes, by taking corresponding samples from the new dataset. In
this case, TP plus FN would lead to haze events, and TN plus FP to non-haze events. The clustering calculations
were done by directly using the k-mean (Steinhaus, 1957) function of scikit-learn library ([https://scikit-](https://scikit-learn.org/stable/modules/clustering.html#clustering)
[learn.org/stable/modules/clustering.html#clustering](https://scikit-learn.org/stable/modules/clustering.html#clustering)). For Beijing cases, the trained machine with 9 features
produced 2591 TP, 11368 TN, 508 FP, and 508 FN outcomes, and 2407 TP, 11484 TN, 492 FP, and 592 FN for
Shanghai. The cluster analysis was performed separately for each of these outcomes in an unsupervised learning
procedure to let the machine to categorize corresponding samples into groups based on similarities among them. In
practice, similarity is judged by the so-called inertia for a cluster with members of $x_i$ and mean of $\mu$:
$$inertia = \sum_{i}^{N}(\|x_i - \mu\|)^2 \qquad (C1)$$
The clustering is to seek a grouping with minimized inertia within each cluster. The overall measure is the
summation inertia that decreases almost exponentially with the increase of number of clusters. In practice, the
cluster analysis was first tested with various given number of clusters ranging from 1 to 100, to examine the values
alongside decay of the inertia. This provided a base to identify the smallest possible number of cluster centers with
reasonably low inertia in actual cluster analysis. This has actually been decided by using square root of the inertia
weighted by the number of samples to put the varying number of samples across various outcomes in consideration.
An optimized number of clusters was chosen with a weighted inertia lower than 1/e of that of the single cluster case.
For TN, due to the large sample number, this criterion was set to be half of 1/e. As a result, the optimized numbers
of clusters for TP, FN, FP, and TN outcomes are 4, 3, 4, and 15 for Beijing and 4, 3 3, and 10 for Shanghai,
respectively,
**(iii)** The members of each cluster derived from (ii) were recorded by the actual sample indices with date
attribute. Therefore, actual samples of input data grouped into various clusters can be thus conveniently identified
with corresponding feature maps retrieved, either in the format of normalized or unnormalized (*i.e.*, in original
quantity as in reanalysis dataset), and used for further analyses. In practice, cluster-averaged maps for various
features were performed beforehand.

## Code and data availability

The Python script for network architecture, training and validation is rather straightforward and simple,
basically consisting of directly adopted function calls from Keras interface library ([https://github.com/keras-](https://github.com/keras-team/keras)
[team/keras](https://github.com/keras-team/keras)) with TensorFlow-GPU ([https://www.tensorflow.org](https://www.tensorflow.org)) as backend, or from scikit learn library
([https://scikit-learn.org/](https://scikit-learn.org/)). All the data used for analyses are publicly available as indicated in the
Acknowledgements.

## Competing interests

The author declares that he has no conflict of interest.

## Acknowledgements

This study is supported by L'Agence National de la Recherche (ANR) of France under "Programme
d'Investissements d'Avenir" (ANR-18-MPGA-003 EUROACE). The author thanks the European Centre for
Medium-range Weather Forecasts for making the ERA5 data publicly available under a license generated and a
service offered by the Copernicus Climate Change Service, and the National Center for Environmental Information
of the US NOAA for making GSOD data available. All the related computations have been accomplished using the
GPU clusters of French Grand equipment national de calcul intensif (GENCI) (Project 101056) and the CNRS
Mesocenter of Computing of CALMIP (Project p18025). Constructive comments and suggestions from two
anonymous reviewers have led to the improvement of the manuscript.

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
