# Peer review of "Forecasting and Identifying the Meteorological and Hydrological Conditions Favoring the 4 Occurrence of Severe Hazes in Beijing and Shanghai using Deep Learning Chien Wang Laboratoire d'Aerologie, CNRS and University Paul Sabatier 14 Avenue Edouard Belin, 31400 Toulouse, France March 2021; Revised A"

_Atmospheric Chemistry and Physics, 2021_

## Author Comment (AC1)

Firstly, I truly appreciate the constructive comments and suggestions from the two reviewers. A statement reflecting this appreciation has been added in the Acknowledgement section. The following are the point-to-point responses to the reviewers' comments (marked with Italic font).

Reviewer #2

*... However, discussions of new scientific findings and more in-depth interpretations are not sufficient. For instance, generally, there is no significant difference in features between clusters (e.g., the four clusters of TP in Beijing look similar), and the author did not explain the physical meanings of the clusters. Thus, it is hard to see the necessity of cluster analysis without significant differences in clusters and explanations of their physical meanings. Although the methods and ideas are innovative, the manuscript is not well-written, mainly that the structure is a bit challenging for readers without sufficient background to follow. The methods could be helpful to the community once the author clarifies and addresses the important issues and includes more scientific discussions about the findings.*

The reviewer's point is well taken. As a result, substantial changes alongside rearrangement of the contents have been made in the revised manuscript, as indicated in the following responses to specific comments. Particularly, the characteristics of each cluster, including the commonality and differences when comparing with others have been better defined and discussed, alongside their meteorological or hydrological meanings aided by necessary elaborations from optical, physical, and dynamical points of view, e.g., why certain identified features could enhance effective sunlight scattering while benefit sustaining hazes, and what are the specific distinctions defining the different regional regimes that influence the cities of interest. In case a further discussion could make the paper seriously exceed an adequate extent, I have also made that clear to the reader, for example, in the summary of clustering section:

"It is worth indicating that the current analysis discussed here is only applied to the included features in clustering, and the presented figures in cluster-wise averaging format might have effectively smoothed out certain variability among members. A full-scale analysis would necessarily go beyond this to provide further synoptical or large-scale hydrological insights and better define different regimes".

I believe these changes in responding to the reviewer's concerns have enhanced the readability of the manuscript while making science findings of the work to be better appreciated.

*Specific comments:*
*1. Page 3, line 95-102: This paragraph is confusing. The first part of the paragraph says that U-net can be used when weather patterns associated with the targeted outcome are known or irrelevant to the task, but the second part mentions that when the environmental conditions related to the targeted outcome are yet known, U-net is not applicable. The two paragraphs are conflicting and confusing. Additionally, the sentences are convoluted and difficult to follow. I suggest rewriting the whole paragraph.*

The sentences have been revised as "In certain applications, the targeted outcomes are the same features as the input but at a different time, *e.g.*, a given weather feature(s) such as temperature or pressure at a given level. The forecasting can thus be proceeded by using pattern-to-pattern

correlation from a sequential training dataset with spatial-information-preserving full CNNs such as U-net …".

*2. Page 4, line 130-132: Are the model structure and parameters the same for the two models (Beijing and Shanghai)? If so, shouldn't the models' parameters be optimized separately for the two regions to obtain the best performance for each of them?*

The two machines were trained and optimized separately, apart from that, many hyperparameters are found to be only sensitive to the general architecture for the two applications using the same features (though with different sizes and from different domains). The sentence has been revised as: "The current version for haze applications of Beijing and Shanghai, though trained separately, contains the same number of parameters of 20,507,161 (11,376 non-trainable) owing to the same optimized kernel sizes".

*3. Page 6, line 187-189: Do you include a testing set (a holdout dataset that has never been used in the training process)? From your results, I think the validation set in this paper is used to tune the hyperparameters of the model and monitor potential overfitting that occurs when training accuracy decreases but validation accuracy increases. An evaluation for a testing set is recommended to assess the performance of the trained model.*

Good question. It has been made much clearly in the revised manuscript as, " For the convenience in comparing performance or restarting training based on a saved machine, a saved training dataset alongside a holdout validation dataset that has never been sued in training, produced following the above procedure, was used".

As indicated in the text that here when to end a training is not determined by the training or testing scores, but a given length (2000 epochs for most cases) to ensure the convergence of the machine performance remarked by a stabilized minimum loss. In practice, cross-validation, multiple ensemble training using different randomized dataset, and repeat/restart training (either partial network or entire network) have all been applied, their results all confirm the convergence of the network performance under the current training procedure as far as a sufficiently long training session is adopted. It is good to use this opportunity to elaborate since the paper is not for discussing these details.

*4. Page 6, line 197-198: Please include the description of class-weight and batch normalization in the appendix or model description and how they help the imbalance-data issue.*

Since the elaboration is rather short and simple, I have it in the text as "…, both using corresponding Keras functions. The class weight is to change the weight of training loss of each class, normally by increasing the weight of the low frequency class. Class weight coefficient was calculated based on the ratio of class 0 to class 1 frequency. Batch normalization (Ioffe and Szegedy, 2015) is an algorithm to renormalize the input distribution at certain step (*e.g.*, each mini batch) to eliminate the shift of such distribution during optimization".

*5. Page 7-8, line 246-249: It is worth checking the maps of features for the TP, FP, and FN for April and May verse the TP, FP, and FN for other months to see their differences in the weather and hydrological features.*

I appreciate the excellent point raised by the reviewer. In fact, one reason for conducting cluster analysis is to seek clear different regimes for various season. As shown in the newly added Table S1 and the (revised) discussions in cluster analysis, the distribution of the number of members within various TP, FN, and FP clusters does not differ evidently in different months (Table S1). This suggests that such an investigate would require an extensive analysis from synoptic and hydrological aspects in a parametric space of cluster-season-outcome, likely involving the use of features that have not been included in clustering. The discussion could easily expand the length of this paper. Nevertheless, in addition to the revised statement in the summary of clustering section cited in the response to the general comments, I have revised corresponding text in the Conclusion as "…an in-depth analysis on weather regimes would necessarily involve the use of certain features that are not included in the current clustering, which, however, exceeds the extent of this paper and can only be discussed properly in a future work".

*6. Page 4-9, Section 2 and 3: The subsections of "Kernel size and CNN performance" and "Reducing the number of input features" could be moved to Section 2, as these two subsections are more related to the model architecture and design. Additionally, the author shows the validation results of reducing the number of input features before introducing the methods of reducing the number of features. The two sections together make it difficult for readers to follow. I suggest reorganizing these two sections and the subsections.*

The reviewer's suggestion has been accepted and the subsection of "Kernel size and CNN performance" has been moved to the Architecture Section with a revised title of "Kernel size optimization". Regarding the feature reduction, this is a content linking the regular machine (16 features) with the new machine with reduced size to be used in, e.g., clustering, and relying on the performance comparison (the result has been included in the Section 3). Although it is a subjective choice, I leave it in the current place.

*7. Page 8, Figure 4: The unit/axis label of top figure?*

Added in caption.

*8. Page 10, line 322: Please specify what VAE stands for.*

The original text "variational autoencoder or VAE" is sufficient to explain VAE, therefore, no change has been made.

*9. Page 11, Section 4: The feature patterns of the clusters for the TP, FN, and FP are very similar with slight differences in certain features. Generally, I don't see the point of conducting cluster analysis because (1) there is no significant difference between clusters, (2) the author did not explain the physical meanings of each cluster (if based on the slight differences), and (3) the author focused on the differences between TP and FN when explaining the missing haze events,*

*which can be done by simply comparing averaged feature maps of TP and FN without cluster analysis.*

The cluster maps with normalized feature (Fig. 7 and others) have been reproduced with rearranged color range plus color bar (thanks to the reviewer's comment later) to better identify the similarity alongside difference among various clusters.

To respond to the reviewer's specific points with summarized revisions (please see the large rewritten Section 4 for details since a copy here would be too long): (1) the largely rewritten discussions in Section 4 have made it more clear that the similarity among TP clusters and one FN cluster as well as two FP clusters is mainly reflected from the local weather and hydrological petterns over Beijing (and Shanghai too) and its immediate surrounding area while the differences are reflected from the airmasses distant from the cities, representing different systems. At the same time, differences among clusters of each outcome from TP, FN, to FP has also been highlighted with necessary details, with characteristic distinctions defining various regimes discussed for different seasons (newly added Table S1) and locations (i.e., Beijing versus Shanghai). (2) The revised discussion has also explained from optical, physical, and dynamical perspective why these common conditions for haze events would favor the occurrence of persistent haze. (3) The new Fig. 7 clearly shows that two FN clusters (with smaller numbers than the first one) differ substantially than TP clusters and the other FN cluster. The revised discussions thus can make a better supported hypothesize that the above difference could be the reason beyond some FN clusters (2 each for Beijing and Shanghai) that haze might occur under an unfavorable condition and lead to the miss forecast by the machine. Obviously, without clustering, this could be overlooked.

*10. Page 11, line 349-354: It seems like the feature patterns of the four clusters for the TP in Beijing cases are very similar (also the clusters for FN and FP). Why did you choose four clusters? Could you justify the purpose of using cluster analysis, given that there is no significant difference between the clusters?*

The number of clusters was determined based on the clustering statistics as described in Methods. As indicated in the opening paragraph of Section 4 that one of the main purposes of clustering here is to better understand the performance of the machine. The reviewer's point towards the similarity of TP cluster should be well addressed in the revised manuscript regarding the differences among TP clusters:

   "Note that each cluster consists of a collection of 3D data volumes or images, any two clusters could be sufficiently differentiated should only one of their images differs based on the clustering derivation algorithm, even though statistically speaking, they very likely belong to the same population (*i.e.*, should be tested statistically). As shown in Fig. 7, the distinctions between TP clusters are largely reflected from the two different airmasses distant from Beijing, in both strength and spatial extent particularly from DTCV patterns, likely representing different types of systems or background regimes. Specifically, a strong DTCV anomalous center seen in cluster 1 and 4 patterns occupies most of the domain west of Beijing and directly influence Beijing and its nearby area. In contrast, DTCV distributions in cluster 2 and 3 are much weaker, where Beijing and its immediate neighboring area even appear to be influence more by the southeaster system. In addition, surface wind distributions of the first two clusters clearly differ from those

of cluster 3 and 4, and the patterns of BLH alongside SW1 and SW2 over Beijing and its immediate neighboring area of cluster3 also suggests a land-atmosphere exchange condition differing from that of others. The combinations of these differences across various TP clusters apparently well defines the various regimes of surrounding weather systems as well as their influence on Beijing. For TP clusters of Shanghai, the above similarities alongside differences among various clusters also exist, except where the cluster 1, 2, and 4 maintain more similarities in feature patterns of distant airmasses from Shanghai, while cluster 3 offers certain evident diversity in many feature patterns comparing to other clusters (Fig. S5). Even more interestingly, the distribution of the number of members within various TP clusters does not differ evidently in different months (Table S1) (note that the number of haze events itself differs seasonally – Fig. 5). Therefore, it is very likely that the characteristic weather conditions favoring haze occurrence and being captured by HazeNet cannot be simply differentiated by locations (Beijing vs. Shanghai) and seasons".

*11. Page 12, line 380: If the results of Shanghai are similar to Beijing and there is no regional characteristic for Shanghai, I suggest removing the results of Shanghai from the paper. Or you could add more discussions and highlight the common or different characteristics shown in the two regions.*

The revised discussions, e.g., by emphasizing on both similarity and difference, contain several extensions to Shanghai cases with necessary descriptions that differentiate the characters of Shanghai cases from those of Beijing. The sentence of "The results of Shanghai are largely the same as in Beijing case (Fig S5 & S6)" have been removed from the revised manuscript.

*12. Page 12, Figure 7: (1) Please include the color bar. The title of the features and cluster labels are hard to see; please enlarge them. It will also be helpful to add a contour map or add a point/shape to label Beijing on the map. (2) As mentioned before, there is no significant difference between clusters, and it is hard to read the key messages from 117 plots, especially that they are all super small. I wonder whether it is necessary to demonstrate the results of all the clusters in the main text? Is it possible to only show the results of the major cluster (the cluster with the largest number) and move the results of all clusters to the supplement? A similar issue also is shown in Figures 8 and 9.*

All suggested changes (color bar, test font, etc.) have been done. The new Fig. 7 'only' include 99 panels now. For an image-recognition based application, I still believe that some readers might be more than interested in comparing the actual images. The point of only showing selected clusters is well taken. However, without, e.g., all FN clusters in Fig. 7, it is difficult to show the difference among them and to address the point of haze might occur under unfavorable conditions.

*13. Page 12, line 386: For the results of unnormalized format, I wonder whether there is a trend shown in the features and the haze events have seasonality or not. If so, how would the trend/seasonality affect the clustered features?*

This indeed was a direction of post-analysis. Nevertheless, as indicated in the previous responses, the immediate result (now added as Table S1) did not provide a clear answer to that

question. Certainly, following the lead of seasonality of haze events, this could be further investigated, though, as the reviewer would also agree that the complexity likely involved makes it a better task for future work.

*14. Page 11, line 349-354 and Page 13, line 398-399: Do the four clusters represent different regimes/scenarios of haze events based on their differences shown in DTCV, SW1, and SW2?*

Very likely as discussed in the revised discussions (with both normalized and unnormalized feature maps). Using normalized maps, it has been made more clearly from U10, DTCV, and BLH. With unnormalized feature amps, more weather regime-alike aspects can be identified and have been discussed. For instance, it is found that Beijing is basically located between two airmass both with anticyclonic surface wind, while for Shanghai these become a northwest anticyclonic and a southeast cyclonic surface wind. These systems, however, appear to be accompanied with different distributions of DTCV and BLH (in some cases even REL). Further investigation would have to involve excluded features such as Z and D, and beyond to offer more synoptical and hydrological insights.

*15. Page 13, line 400 and line 402: It should be Figure S7 and Figure S8?*

Corrected.

---

## Author Comment (AC2)

Firstly, I truly appreciate the constructive comments and suggestions from the two reviewers. A statement reflecting this appreciation has been added in the Acknowledgement section of the revised manuscript. The following are the point-to-point responses to the reviewers' comments (marked with Italic font).

Reviewer #1

*Specific comments*
*1. The selected threshold for defining a severe haze event in the 2-class training is set to days whose surface visibility decreases below the 25 While the value of that percentile varies in time and space, I suggest elaborating more on selecting that exact percentile threshold.*

The point is well taken, the corresponding text has been revised to "Although p25 values vary interannually, their long-term means actually represent…".

*2. Please elaborate on the input data and its possible effects on the results. How many ground stations are used for the analysis in each city? What are the potential limitations that are resulted from the ERA5 spatial resolution of 0.25 degrees?*

I assume this question matters both label and input. For labeling data, this has been made more clearly by adding "…observations in corresponding airports of these two cities during during the time from 1979 to 2019…". The input data obtained from much larger domains to reflect regional weather and hydrological conditions used for forecasting the occurrence of haze at locations of interest. The grid numbers have been provided in the original manuscript already, as in Line 176-177 "…Beijing (64x96 grids) and Shanghai (64x64 grids)…". On the resolution, the discussions of kernel size (please note that this has been moved to the Section 2 in the revised manuscript) alongside highlights in both Abstract and Conclusion have provided insights on that, i.e., the machine actually prefers feature patterns in a larger scale (5-6 degree) than a smaller one in performing the forecast, therefore, the 0.25 degree resolution of ERA5 is adequate for the purpose.

*3. Please include, possibly as a supplemental, some technical details regarding the CNN analysis. What were the data and computational volumes and costs? What kind of computation platform was used, how long each training session take? What were optimization and approximation procedures implemented? Such information could assist future researchers when planning their analyses and also provide the scientific community with a technological benchmark for comparison with future projects.*

A brief description has been added to the end of Section 2 as "Entire trainings have been conducted using a NVIDIA Tesla V100-SXM2 GPU cluster, costing 25s and 17s per epoch for the machine of Beijing and Shanghai, respectively". The paper also highlights the optimization of an important hyperparameter for meteorological applications, *i.e.*, the kernel size of first two CNN layers. In addition, text also added to explain algorithms such as class-weight and batch normalization. Discussions on other "routine" procedures are omitted or could be referred to Wang (2020, arXiv) to limit the paper size within a reasonable length.

*Technical corrections*
*1. Please keep consistency in number representation along the manuscript (e.g. "11,376" in P. 4 Line 132 vs. "14975" in P. 11 Line 338.*

Done.

*2. Please keep consistency in technical terminology along the manuscript (e.g., "class-1" vs. "class 1", etc.).*

Done.

*3. Please specify the unit following physical quantities (e.g. "...heights at 500 (Z500) and 850 (Z850) hPa" in P. 6 Line 188).*

Good point, the unit of geopotential height is in meter, here hPa is the unit for pressure levels. It seems to me that the meaning should be quite clear. In a later part of the cluster analysis using non-normalized quantities marked by color bars (e.g., Fig. 8 caption), all the units are indeed provided.

*4. Please follow a consistent terminology for classes 0 and 1 in the 2-class analysis (e.g., "non-haze events" and "severe haze events").*

The point is well taken. I have checked the text rather thoroughly to make sure class 1 and class 0 are only used when classification is concerned while the other for event-based discussions.

*5. 2 Line 38: Please change "event" to "events".*

Done.

*6. 2 Line 39: Please change "has" to "has".*

I believe the reviewer meant change "has" to "have", if so, done.

*7. 4 Line 119: Please change "Introduction" to "introduction".*

Done.

*8. 5 Line 158: Please add punctuation marks where necessary.*

Done.

*9. 5 Line 176: I suggest replacing or dropping the words "longitude-latitude" that are already self-embedded in surface map objects.*

Done.

*10. 6 Line 214: Please change "metrics" to "metric".*

Metrics is adequate here for the purpose.

*11. 6 Lines 219-220: Please clarify that sentence.*

The sentence has been revised to "…a validation accuracy of 80% (frequency of non-haze events or no-skill forecasting accuracy) in both…".

*12. 7 Line 225: Please correct a typo "class0weight"*

Done.

*13. 8 Line 267: Please modify to "there are many hyper-parameters in HazeNet that need …".*

I believe the reviewer was referring to Line 257. A sentence has been added there as "As in the cases of other CNNs, there are many hyperparameters need to be determined or optimized. These have been done through numerous testing trainings. In practice, it occurs that,…".

*14. 12 Line 398: Please change "soli" to "soil".*

Done.

*15. 8 – caption (Lines 405-408): Please specify the explicit variable descriptions for better readability.*

Done.

---

## Referee Report (RR1)

The author has addressed nearly all of my previous comments and revised the manuscript. The quality of the manuscript has improved a lot, particularly the interpretation of the cluster analysis. The revised manuscript indeed provides a more comprehensive understanding of how the neural network classifies haze and non-haze events over Beijing and Shanghai. Only minor clarification and revision remain, as stated below:

Specific comments:
1. Page 2, abstract: The abstract reads like a short summary of the introduction. I suggest the author includes some key findings (e.g., the performance of the model and/or the identified haze-favorable environment in Beijing and Shanghai) in the abstract.

2. Page 4-7, Section 2: I still feel that the structure of this section is quite complex for readers to follow. I think it is better to add subsection titles and rearrange paragraphs a little bit. The subsections can be defined as follows: 2.1 network architecture (including content of lines 128-154), 2.2 kernel size optimization, 2.3 Data (lines 155-191), and 2.4 Training methodology (lines 192-214). The subsection bullet points (e.g., 2.1) may not be necessary but the author could at least have titles bolded like what you already had for kernel size optimization.

3. Page 7, line 232: "…. see next section and Method)…." I am not sure about which "Method" you refer to here.

4. Page 9, line 301: The author has a subtitle for this section for reducing input features, but there is no subtitle for the prior paragraphs. This is quite confusing for readers to understand the number of input features you used in the results presented in Figures 4 and 5. I suggest the author add bold subtitles for the prior paragraphs: "Model performance using 16 input features".

5. Page 10, lines 321-322: It seems like the following cluster analysis uses the model results with nine input features. Maybe the author can rewrite or add another sentence clearly stating that the subsequent cluster analysis is conducted using the model outputs with nine input features.

6. Page 12, lines 412-416: It will be helpful if the author could provide more analyses, discussions, or insights on why HazeNet misses the FN cases, especially the cases in cluster 1 since they are major cases (Table S1). It seems to me that the key differences between cluster 1 in FN cases and TP cases are shown in U10, V10, DTCV, and SW1. Do the patterns of cluster 1 (or other clusters) in FN cases represent weather patterns unfavorable for haze events?

7. Page 14, lines 438-442: In my previous comment #13 I asked how the trend or seasonality would affect the clustered features. I agree that the analysis could be for future work, but I suggest the author includes several sentences mentioning this point and potential influences of trend or seasonality on the clustered results.

---

## Author Response (AR2)

Again, I truly appreciate the constructive suggestions from the reviewers. The following are the point-to-point responses to the reviewers' comments (marked with Italic font).

Reviewer #2

*Specific comments:*
*1. Page 2, abstract: The abstract reads like a short summary of the introduction. I suggest the author includes some key findings (e.g., the performance of the model and/or the identified haze favorable environment in Beijing and Shanghai) in the abstract.*

The reviewer's suggest has been accepted. The revised Abstract should read more specific on the findings of this study:

"Severe haze or low visibility event caused by abundant atmospheric aerosols has become a serious environmental issue in many countries. A framework based on deep convolutional neural networks containing more than 20 million parameters, namely HazeNet, has been developed to forecast the occurrence of such events in two Asian megacities: Beijing and Shanghai. Trained using time sequential regional maps of up to 16 meteorological and hydrological variables alongside surface visibility data over the past 41 years, the machine has achieved a good overall performance in identifying the haze versus non-haze events and thus their respectively favorite meteorological and hydrological conditions, with a validation accuracy of 80% in both Beijing and Shanghai cases, exceeding the frequency of non-haze events or no-skill forecasting accuracy, and a F1 score specifically for haze events nearly 0.5. Its performance is clearly better during months with high haze frequency, that is all months except dusty April and May in Beijing and from late autumn through entire winter in Shanghai. Certain valuable knowledge has also obtained from the training such as the sensitivity of the machine's performance to the spatial scale of feature patterns that could benefit future applications using meteorological and hydrological data. Furthermore, an unsupervised cluster analysis using features with a greatly reduced dimensionality produced by the trained HazeNet has, arguably for the first time, successfully categorized typical regional meteorological-hydrological regimes alongside local quantities respectively associated with haze and non-haze events in the two targeted cities, providing substantial insights to advance our understandings of this environmental extreme. Interesting similarities in associated weather and hydrological regimes between haze and false alarm clusters, or differences between haze and missing forecasting clusters have also been revealed, implying that factors such as energy consumption variations, long-range aerosol transport, and beyond could also influence the occurrence of hazes, even under unfavorite weather conditions".
.
*2. Page 4-7, Section 2: I still feel that the structure of this section is quite complex for readers to follow. I think it is better to add subsection titles and rearrange paragraphs a little bit. The subsections can be defined as follows: 2.1 network architecture (including content of lines 128-154), 2.2 kernel size optimization, 2.3 Data (lines 155-191), and 2.4 Training methodology (lines 192-214). The subsection bullet points (e.g., 2.1) may not be necessary but the author could at least have titles bolded like what you already had for kernel size optimization.*

This is a good point. Three subsections have been created, respectively with titles of: 2.1 Network architecture; 2.2 Training data and methodology; and 2.3 Kernel size optimization.

*3. Page 7, line 232: ".... see next section and Method)...." I am not sure about which "Method" you refer to here.*

This perhaps is a leftover from an older version. The words of "see next section and Method" have been removed.

*4. Page 9, line 301: The author has a subtitle for this section for reducing input features, but there is no subtitle for the prior paragraphs. This is quite confusing for readers to understand the number of input features you used in the results presented in Figures 4 and 5. I suggest the author add bold subtitles for the prior paragraphs: "Model performance using 16 input features".*

The reviewer's point is well taken. In the captions of Fig. 4 and 5, the number of features used in training were indicated in the previous version. To avoid further confusion, an additional note of "with 16 features" has been added in Fig. 4 caption on original Line 257 for (Right Top) panel. In addition, a sentence was added in the end of the opening paragraph (original Line 252): "Also note that, unless otherwise indicated, results shown in this Section are obtained using 16 features". Therefore, the only exception of the above is in Fig. 4 (right Bottom), where a note of "16 and 9 features" already existed (so did the legend notes of the figure). Lastly, "(Fig. 4)" in the Line 321 (original) has been revised to "(Fig. 4, Right Bottom)".

*5. Page 10, lines 321-322: It seems like the following cluster analysis uses the model results with nine input features. Maybe the author can rewrite or add another sentence clearly stating that the subsequent cluster analysis is conducted using the model outputs with nine input features.*

Yes, indeed a sentence is needed here. It has been added in the original Line 340-341, "…the trained HazeNet for Beijing and Shanghai using 9 instead of 16 features, benefited from the effort of reducing the number of input features as described in the end of last Section, have been used here to…".

*6. Page 12, lines 412-416: It will be helpful if the author could provide more analyses, discussions, or insights on why HazeNet misses the FN cases, especially the cases in cluster 1 since they are major cases (Table S1). It seems to me that the key differences between cluster 1 in FN cases and TP cases are shown in U10, V10, DTCV, and SW1. Do the patterns of cluster 1 (or other clusters) in FN cases represent weather patterns unfavorable for haze events?*

The original discussions from Line 409 to 416 have been revised as: "On the other hand, among three FN clusters (also associated with haze events but missed in prediction), only the first cluster (the major cluster of FN) displays certain similarity to TP clusters across various features. Even for this cluster, the characters of the airmasses distantly surrounding Beijing differ substantially from those of TP clusters, as seen from the patterns of temperature (DT2M, T2SM), wind particularly V10, and column water (DTCV) that reflect a much weaker weather system on the west. The patterns of BLH, SW1, and SW2 also differ from those of TP, indicating a different near site boundary layer and hydrological condition. Such differences appear to be even more evidently in the two other (minor) clusters, *e.g.*, the size and strength of high relative

humidity center covering Beijing are even different. This result suggests a possible reason for why HazeNet missed these targets, that is haze might occur under unfavorable weather and hydrological conditions owing to, *e.g.*, certain energy consumption scenarios".

*7. Page 14, lines 438-442: In my previous comment #13 I asked how the trend or seasonality would affect the clustered features. I agree that the analysis could be for future work, but I suggest the author includes several sentences mentioning this point and potential influences of trend or seasonality on the clustered results.*

The original sentence of "…to appreciate the conventional regional and local meteorological and hydrological patterns associated with various events" has been revised to "…to appreciate the conventional regional and local meteorological and hydrological patterns, and to detect thus to implement additional analysis, if necessary, on the possible impact of seasonality or trend associated with various events".